# Exploring the Smart Street Management and Control Platform from the Perspective of Sustainability: A Study of Five Typical Chinese Cities

**Fanding Xiang, Haomiao Cheng \***  **and Yi Wang**

College of Architecture and Urban Planning, Beijing University of Technology, Beijing 100021, China
* Correspondence: chenghaomiao@bjut.edu.cn

**Abstract:** In the context of "smart cities" and stock-based development, street renewal focuses more on quality and sustainability in China. To improve the efficiency of current smart technology applications, a comprehensive whole-life cycle system needs to be established in street space. After sorting out the application of smart technologies in the street design guidelines (SDGs) for typical cities in China, the compilation and application of smart technologies for sustainability were categorized into five areas: smart transportation, convenient living, life enrichment, the protection of vulnerable people, and environmental monitoring. Based on theoretical support and realistic needs, a smart street management and control platform (SSMCP) was built. The SSMCP is divided into four layers: the basic information layer for the background, the technology platform layer for the core processing, the institutional protection layer for the guarantee mechanism, and the scene application layer for spatial interactions. The results can provide a scientific reference for improving the sustainability of street space and implementing a "smart cities" project at the street level.

**Keywords:** smart city; sustainable design; stock-based renewal; street design guidelines; China

## 1. Introduction

The evolution of the digital age brought about changes in smart technologies, giving rise to efficient development in many fields worldwide, such as finance, the military, ecology and environment, social and economic life, and so on [1]. In the field of urban construction, the promotion of smart cities through smart technologies has broadened its scope of application. Smart cities apply computing technology in urban planning and construction management, such as cloud computing, big data, and spatial geographic information, which make urban infrastructure more interconnected and efficient, while empowering the government with efficient operation and management mechanisms, as well as providing better living services for people [2]. As constructing a "smart city" is a significant strategic opportunity, smart construction, the smart coordination of resources, and the smart management of data has advanced rapidly in China [3]. In December 2015, the China Central Urban Work Conference noted that urban renewal should conform to the "new normal", adhere to the concepts of "smart growth" and "stock-based renewal", and promote the transformation of urban development for connotative growth [4,5]. Smart technologies should be focused on actively promoting the sustainable construction of urban public spaces in relation to the aspects of health, safety, and livability [6,7]. Therefore, the mode of enhancing the competitiveness of cities through smart technologies and sustainable development has gradually reached a consensus [8]. Applying smart technologies to the redesigning of urban space has become possible [9].

The use of smart technologies to realize stock-based development is gradually being explored in China. The city information model (CIM) has been developed to serve as the basis of smart construction [10,11]. The data-control platform elements present the

characteristics of a giant system with multiple objects, departments, and levels [12]. Currently, there are several problems, such as urban data silos, single-data scenarios, a lack of information technology [13], and unsound operation and management systems [14]. In terms of street space, the application of smart technologies lacks theoretical support and overall control of the whole-life cycle, and there is insufficient consideration of the concepts of overall management and control, planning, and co-governance construction.

The city is a complex mega-system, with many sectors involved in the design of street space [15]. Street space contains many elements, and street space management involves many construction departments, as its functions are comprehensive and complex [16]. Research on developing smart street space in China began in 2016. Shanghai issued the first city-level SDGs, which answered the questions: "what kind of sustainable streets should be built?" and "what kind of smart technology should be applied?". Since then, several SDGs have been successively compiled and have continuously enriched the connotations of street space according to local circumstances (Figure 1). However, the overall coordination of the system of traffic, municipal pipelines, landscape, and urban furniture is unclear. As the SDGs provide more explicit valuable concepts and mature technical support, using smart technologies to build a comprehensive system has become an effective way to achieve sustainable development. Promoting street renewal requires a robust and comprehensive platform for the overall planning, design, and management of street space [17,18].

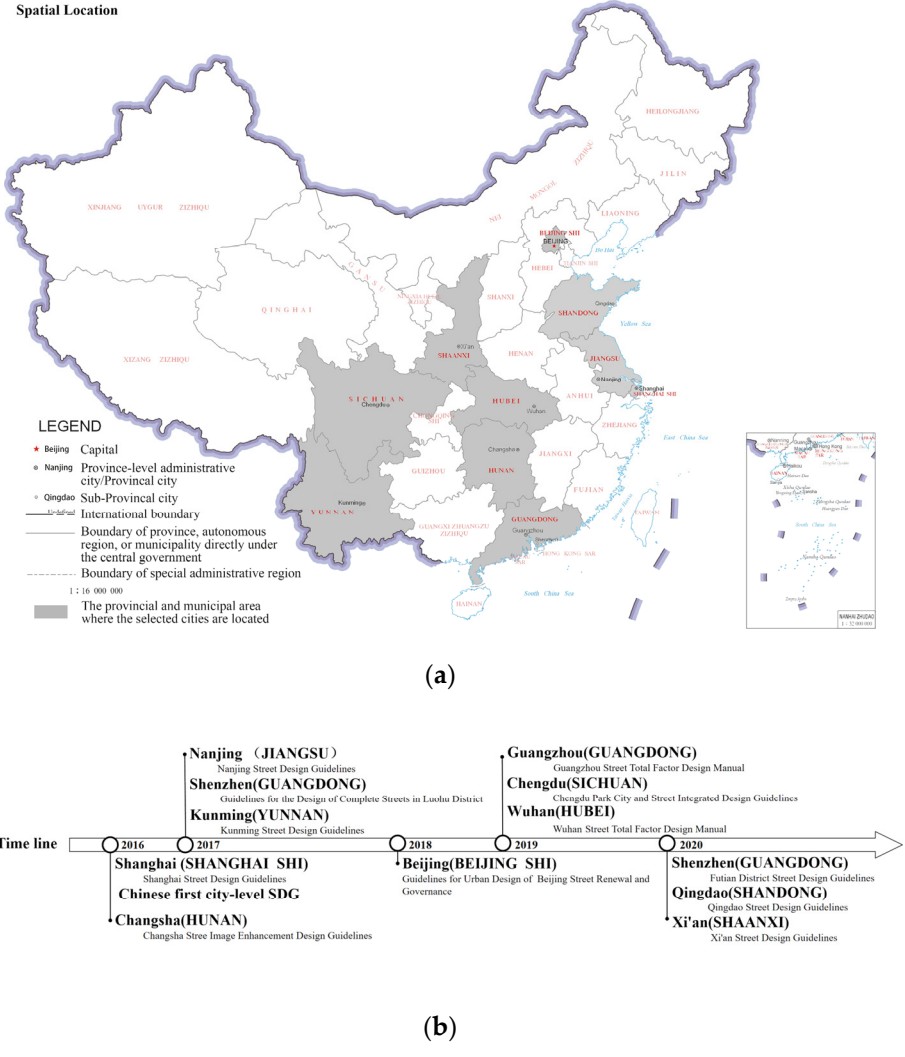

**Figure 1.** Spatial locations and timeline of SDGs in China ((**a**): spatial location; (**b**): timeline). Source: by author on the basis of data from [19–24].

In the context of smart cities, the use of smart technologies is becoming more widespread. Following the achievements of China's smart city construction and stock-based renewal, this study demonstrated the validity and necessity of improving sustainability and quality in street space based on analyzing SDG data. The SDG data obtained in this paper allowed us to sort out the types, ranges, and methods of application of smart technologies in each selected case. Meanwhile, in the process of analyzing data, some insightful details were highlighted as figures. After comprehensive comparison, a summary of the applications of various smart technologies, which was the theoretical basis as well, was created to present the scenarios of application and technical equipment of SDGs. Based on those details and the theoretical basis, a comprehensive platform was explored, which was the Smart Street Management and Control Platform (SSMCP) (Figure in Section 3.3). This research aimed to show an integrated real-case framework for a whole-life-cycle system based on street space, which can be of significance for smart cities research in China in the future.

## 2. Methods and Data

### 2.1. Experimental Methods

The literature review method was used in this study to analyze the SDGs of typical cities in China, and a qualitative comparative analysis method was used to build the SSMCP framework. The literature review method is a systematic way of locating and analyzing arguments. It can aid in formulating search strategies for different databases, conducting systematic studies on a particular issue, and drawing conclusions. The literature review method comprises four steps: question raising, literature determination, data extraction, and presentation of conclusions [25]. In this paper, a sustainable perspective on smart technology was proposed, and the retrieval object was determined to be the SDGs of typical cities in China. The focal points of the SDGs were summarized via text sorting. A qualitative comparative analysis method was then used to identify shared attributes in the scientific information presented in the literature. Based on the essential differences between different city development visions, the wholes and the parts that need to be compared in each city were identified, and the sustainable development and smart technology applications were summarized. This method ensured that the path of the comparative analysis could distinguish the focuses and commonalities and the underlying causal logic could be explored [26]. Based on the literature review and comparative analysis, the framework of the SSMCP was proposed.

### 2.2. Experimental Data

The present study analyzed the SDGs of five typical cities in China. SDGs are specialized and systematic technical manuals and methods for guiding construction and design [27]. Each of the five typical cities is endowed with unique themes and characteristics (Table 1). As China's capital city, Beijing assumes the function of a window onto city life. To optimize the elements of street space, the SDGs put forward the requirements for sustainable development as their orientation, and they highlight the value of delicate design. Value transformation emphasizes a people-oriented priority; holistic management and control; and diversified collaboration, coordination, and overall planning [28]. Shanghai is envisioned as actively responding to new urban construction and building a modern city that is harmonious, livable, vibrant, and distinctive. It also advances the construction goals of "prosperity and innovation, health and ecology, happiness and humanity", which lend a focus to transforming the mode of urban development and achieving endogenous growth through organic renewal. To further implement harmonious and livable spaces, the guidelines emphasize strengthening street design, improving service supply, and shaping the city spirit [29]. Shenzhen promotes the general theme that "the core of the city is people". There, urban development is expected to adhere to the principle of moving from "Shenzhen speed" to "Shenzhen quality". Their guidelines indicate that urban renewal and ecological restoration should be carried out on the micro-level of street space to improve urban quality. The development goal of "safety, vitality, beauty, wisdom, and

green" is supported by smart facility planning and design [30]. Against the background of stock-based development, Nanjing is committed to promoting the transformation of urban development through urban design. The guidelines emphasize convenience and a sense of the scale of streets as standards for measuring the degree of perfection. Focusing on the goal of "building a modern international humanistic and green city", the guidelines highlight a "green, humanistic, smart and intensive" orientation [31]. Qingdao emphasizes that the human living environment is the intrinsic driving force of urban development. The guidelines introduce the four concepts of "people-oriented, spatial coordination, organic integration, and system coordination", which create an engine for sustainable street-level development.

**Table 1.** Overview of the five typical cities. Source: by author on the basis of data from [19–24].

| City | Level | Location | Vision |
|---|---|---|---|
| Beijing | Capital city | Northern China | Harmonious City of Sustainability |
| Shanghai | Province-level administrative city | Eastern China | Prosperity, Health, and Happiness |
| Shenzhen | | Southern China | From Speed to Quality |
| Nanjing | Provincial city | Eastern China | Modern International Green City of Humanities |
| Qingdao | Sub-provincial city | | Humanization Design |

## 3. Discussion and Results

### 3.1. Selected Case Studies on Smart Technology Application

The sorting of the selected cases was conducted, and all the smart technologies and application scenarios were recorded. Some especially insightful applications from selected cases were highlighted with figures.

### 3.1.1. Beijing

*The Guidelines for the Urban Design of Beijing Street Renewal and Governance* promote the development of technology applications from the perspective of efficiency and sustainability (Table 2) [19]. The guidelines propose that technologies are used for smart transportation, and they advocate that signal light poles should hold some electronic equipment (Figure 2a) and be integrated to save space. The guidelines present a vision of multifunctional combinations of urban furniture, allowing that furniture to provide a more comprehensive range of convenience services, such as newsstands (Figure 2b). In addition, from the perspective of environmental monitoring, sensors on the streetlamp shades could monitor the microclimate in real time by collecting various types of data, such as on air pollutants, light intensity, noise, heat islands, etc. The sensors could also monitor the flow of people and calculate signaling data to assess street vitality, ultimately giving feedback to managers via wireless networks. The guidelines also suggest that street-level data could be shared and first-hand information could be used for terminal analyses.

**Table 2.** The applications of smart technologies in the Beijing SDGs. Source: by author on the basis of data from [19].

| Application Scenario | Object | Purpose | Description |
|---|---|---|---|
| Smart Transportation | Signal Light Pole | Collect traffic data | The poles carry sensors for traffic flow detection and road hazard detection |
| | Bus Stop | Provide bus location information | Build a bus information platform using big data to provide bus arrival information |
| | Vehicle Lane | Improve driving efficiency | Form a green-wave traffic zone via traffic-light signals and dynamically add reversible lanes |

**Table 2.** *Cont.*

| Application Scenario | Object | Purpose | Description |
|---|---|---|---|
| | Shared Bicycle | Increase utilization | Real-time control and regulation of bicycle location and use through apps |
| | Parking Lot | Optimize parking resources | Build a parking-fee system to realize parking-space sharing |
| | Comprehensive Platform | Improve urban efficiency | Use of terminal data analysis for electronic warnings |
| Convenient Living | Public Art Installations | Enhance interactions | Expand communications media, such as images, sounds, smells, and tactile experiences through art installations |
| | Urban Furniture | Provide self-service facilities | Promote the installation of interactive information systems in facilities such as newsstands, bus stops, and garbage bins to provide retail, Wi-Fi, charging piles, and other services |
| | Smart Device | Information sharing | Information interaction between apps, parking cloud platforms, delivery services, etc. |
| Environmental Monitoring | Streetlamp Shade | Collect environmental data | Monitor the local climate environment via timed and photoelectric control equipment |

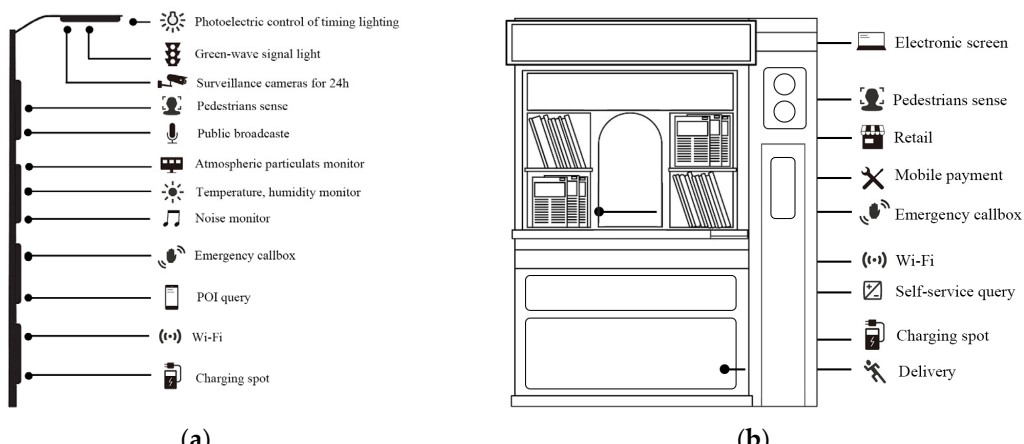

(**a**)       (**b**)

**Figure 2.** Urban furniture design recommended in the Beijing SDGs ((**a**): light pole; (**b**): newsstand). Source: by author on the basis of data from [19].

### 3.1.2. Shanghai

*The Shanghai Street Design Guidelines* state that new materials and technologies applied in the street space should achieve sustainable development [20]. The sustainable construction of streets according to the guidelines can be summarized as relating to five aspects: transportation, life, vitality and enrichment, safety, and the environment (Table 3). The guidelines propose that the coordination of transportation facilities is more efficient in facilitating residents' mobility. As relates to the aspect of convenient living, the installation of electronic-screen-realized multisource information dissemination and real-time release of various city information is recommended. The guidelines also state that the consolidation of municipal facilities into a facility belt should be encouraged (Figure 3). As an improvement for life enrichment, some communications media were incorporated into public art installations. The application of audio, video, and heat-sensing technologies could improve self-protection. The use of interactive media, data terminals in urban furniture, and multiple sensors achieve monitoring and management of the living environment. The guidelines also propose that data should be collected and feedback provided so as to

dynamically adjust urban activities via various smart technologies. The establishment of a comprehensive smart city platform is also proposed to analyze different activities.

**Table 3.** The applications of smart technologies in the Shanghai SDGs. Source: by author on the basis of data from [20].

| Application Scenario | Object | Purpose | Description |
|---|---|---|---|
| Smart Transportation | Signal Light | Improve traffic efficiency | Create a green-wave traffic belt and establish a bus-only signal system |
| | Bus Stop | Provide bus information | Make electronic station signs and provide an outlet for passenger complaints and other services |
| | Shared Bicycle | Combined with public transportation system | Obtain information on available bicycles through the public transportation system and make reservations for borrowing and returning bicycles |
| | Parking Lot | Optimize parking resources | Establish a parking guidance and parking-space-sensing system |
| | Traffic Information Panel | Improve information coverage | Set up information terminals that can display all kinds of traffic information and reduce dependence on mobile phones |
| Convenient Living | Electronic Screen | Provide handy information | Use screens to provide information for daily life, business, and medical care and to display security and disaster warning information |
| | Newsstand | Provide life services | Provide self-service retailing, charging piles, Wi-Fi, express delivery, mobile payment, and other services |
| | Garbage Can | Reduce pollution | Use solar energy to compress the volume of garbage, notify sanitation personnel of the transfer, and provide recycling information |
| | Municipal Facility | Intensify space | Encourage "multipurpose for one pole and box" and control the occupied proportion of facilities |
| Life Enrichment | Public Art Installations | Increase street vitality | Expand communications media, such as images, sounds, smells, and tactile experiences through art installations |
| Protection of Vulnerable People | Audio and Video Surveillance Equipment | Maintain security | Establish an analytical platform to automatically identify special situations and establish an early warning system for natural disasters via audio, video, and heat-sensing technologies |
| | Emergency Callbox | Focus on the needs of vulnerable people | Provide signal sound alerts at intersections and set infrared sensor alert devices at pedestrian crossings |
| Environmental Monitoring | Streetlamp Shade | Collect environmental data | Load with sensors for the real-time monitoring of noise, air quality, and temperature |
| | Green Irrigation System | Save water | Dynamic adjustment of irrigation time and volume through humidity sensing |

### 3.1.3. Shenzhen

The application of smart technologies in Shenzhen's SDGs is reflected in smart transportation, convenient living, the protection of vulnerable people, and environmental monitoring (Table 4) [21,22]. In terms of smart transportation, the guidelines state that technologies such as radar and geomagnetic induction could be used in signal light poles to record the spatial and temporal characteristics of people and vehicles in traffic. In regard to convenient living, the guidelines propose the concept of a smart life micro-hub, which could provide customized demand services (Figure 4). Considering vulnerable people, the feasibility of providing protection could be enhanced through the application of infrared thermal imaging facilities and sound devices. Regarding environmental monitoring, the

guidelines recommend comprehensive monitoring via environmental sensors, and data density in key areas should be strengthened.

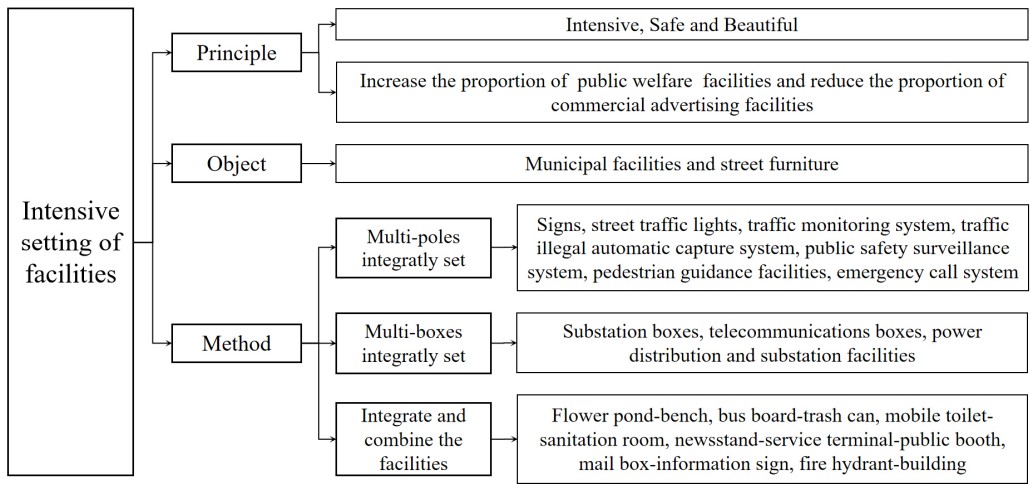

**Figure 3.** The integration modes of street space noted in the Shanghai SDGs. Source: by author on the basis of data from [20].

**Table 4.** The applications of smart technologies in the Shenzhen SDGs. Source: by author on the basis of data from [21,22].

| Application Scenario | Object | Purpose | Description |
|---|---|---|---|
| Smart Transportation | Signal Light Pole | Record human and vehicle data | Use radar, geomagnetic and thermal sensing, satellite positioning, IoT, and other technologies to record information on the flow of people and the type and number of vehicles |
| | Composite Transportation System | Establish a barrier-free travel system | Customize travel needs for elderly, sick, disabled, and pregnant people, and implement the overall design in conjunction with barrier-free facilities |
| | Traffic Information Panel | Provide traffic information | Build a comprehensive traffic search panel and provide bus, subway, train, plane, and ferry information |
| Convenient Living | Smart Life Micro-Hub | Improve work efficiency | Use smart life micro-hubs to customize the shift-level connection of life services for office workers based on travel demands and to coordinate the connection of office, shopping, and other activities with transportation information |
| | Urban Furniture | Provide life services | Newsstand equipped with charging, Wi-Fi, shopping, and other functions |
| Protection of Vulnerable People | Underpass | Reduce crime rates | Provide a responsive space in urban underpasses with lights and sounds to improve safety |
| | Safety Devices | Improve safety at street crossings | Use infrared thermal imaging facilities to monitor the trajectory of pedestrians; add ground signals and intelligent road studs to ensure pedestrian safety |
| | Sound Devices | Protect the visually impaired | Visually impaired people can identify the signal by sound, and the volume is automatically adjusted according to the ambient noise |
| Environmental Monitoring | Streetlamp Shade | Collect environmental data | Monitoring of air pollutant data, noise, temperature, humidity, wind speed, and key pollution sources |

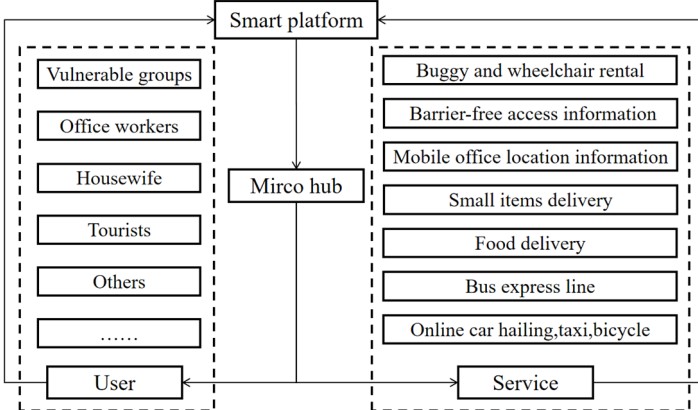

**Figure 4.** Smart life micro-hub system created in the Shenzhen SDGs. Source: by author on the basis of data from [21,22].

### 3.1.4. Nanjing

*The Nanjing Street Design Guidelines* apply smart technologies to the areas of smart transportation, convenient living, and the protection of vulnerable people (Table 5) [23]. They also present the idea of an information platform that could share data on these application scenarios (Figure 5). The guidelines propose transportation solutions, such as a bus corridor, green-wave transportation, a parking guidance system, and transportation hubs to improve the efficiency of urban transportation. As relates to convenient living, urban furniture should be designed to achieve sustainable development functions, such as energy saving, a low carbon output, and self-sensing using loading sensors. Furthermore, the guidelines present methods to improve safety at street crossings for vulnerable people by implementing audio and infrared induction prompters.

**Table 5.** The applications of smart technologies in the Nanjing SDGs. Source: by author on the basis of data from [23].

| Application Scenario | Object | Purpose | Description |
|---|---|---|---|
| Smart Transportation | Bus Corridor | Improve the efficiency of public transportation | Allocate bus corridors on main traffic roads and establish bus-only signal systems |
| | Traffic Surveillance System | Collect traffic data | Set up traffic monitoring facilities near road intersections to achieve the comprehensive management of traffic flow |
| Convenient Living | Streetlamp | Save energy | Encourage the application of inductive sidewalk streetlights to provide targeted lighting |
| | Bus Stop | Provide weather information | Bus stops display weather forecasts and provide travel guidance |
| | Newsstand | Provide life services | The newsstand introduces multimedia data terminals to accept queries and provide street and surrounding information, and it is equipped with Wi-Fi, transitioning to media information terminals |
| Protection of Vulnerable People | Signal Light | Improve safety at street crossings | Add signal-light sound prompts, infrared induction prompting devices, and rescue facilities for vulnerable people |

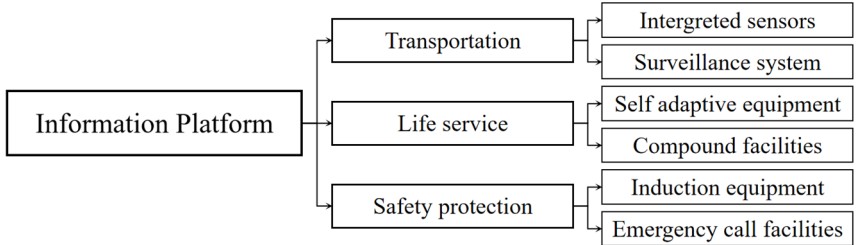

**Figure 5.** The framework of the information platform presented in the Nanjing SDGs. Source: by author on the basis of data from [23].

### 3.1.5. Qingdao

*The Qingdao Street Design Guidelines* propose a method for extracting urban information using smart technologies to promote street management's transformation toward a more sustainable orientation (Table 6) [24].

**Table 6.** The applications of smart technologies in the Qingdao SDGs. Source: by author on the basis of data from [24].

| Application Scenario | Object | Purpose | Description |
|---|---|---|---|
| Smart Transportation | Intelligent Cloud Computing Platform | Improve traffic efficiency | According to the human and vehicle flow data, the dispatching of buses and taxis, online car hailing, rail transit, and static parking can be carried out by the intelligent cloud computing platform |
| | Signal Light | Provide green lanes for special vehicles | In the event of an emergency, the signal light uses traffic flow data to automatically allocate time to customize green lanes for ambulances and fire engines |
| Convenient Living | Oblique Photography Technology | Collect information on the physical spaces of streets | Oblique photography technology can be used to collect street morphology and color data for analysis of landscape corridors and city skylines to create a higher quality of life |
| | Portrait Technology | Improve business vitality | POI data can be used to analyze the advantages of street businesses and business models and to analyze the characteristics of the crowd for portrait technology, so as to match to commercial business and stimulate consumption |
| | Electronic Information Screen | Provide information for queries | Electronic information screens can provide all types of life information |
| | Newsstand | Provide life service | Newsstands can add self-service retail, charging piles, and express services |
| Protection of Vulnerable People | High-density Sensors | Optimize information dissemination channels | High-density urban data sensors can be used to perceive changes in the city's micro-environment, to predict future spatial and temporal development trends, and to establish an early warning information system to warn the city of accidents, disasters, and public health emergencies |

The guidelines propose that traffic scheduling should be carried out based on the spatial–temporal characteristics of human and vehicle flows and should be supported by an intelligent cloud computing platform. As to convenient living, the guidelines point out that technologies such as oblique photography and portrait technology should be used to collect basic urban information (Figure 6). The guidelines also promote the design of smart electronic screens, and they advocate a combined functional structure for urban furniture to

improve life services. In consideration of vulnerable people, high-density coverage sensors were proposed so as to optimize information dissemination channels.

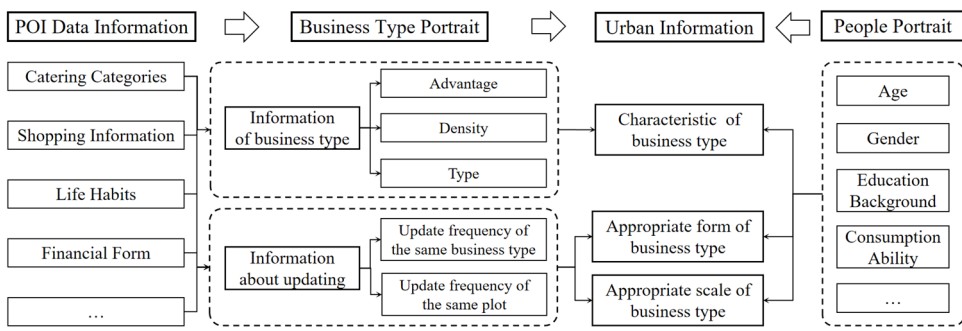

**Figure 6.** Business activity measurement portrait system in Qingdao. Source: by author on the basis of data from [24].

### 3.2. Summary of Various Smart Technologies in SDGs

In summary, the applications of smart technologies to achieve sustainable development in each of the SDGs mainly concentrate on five aspects: travel and transportation information, the convenient integration of residents' lives, improvement of the vitality of public facilities, protection of disadvantaged groups, and real-time monitoring of the living environment (Table 7). The raw data of selected cities on the application of smart technologies in the SDGs are shown in Tables S1–S5 in the Supplementary Material.

**Table 7.** Summary of smart technologies for sustainable development in the SDGs. Source: By author.

| Application Scenario | Concept | Purpose | Technical Equipment |
|---|---|---|---|
| Smart Transportation | To create stable transportation and intermodal hubs | Creating a green-wave traffic belt for traffic-flow dispatching. Sharing static traffic space and forecasting traffic data | GPS, electromagnetic induction devices, electronic touchscreen technology, big data, app terminal, IoT |
| Convenient Living | To disseminate urban information | Integrating urban information and public resources | Wi-Fi, unmanned self-service system, smart space |
| Life Enrichment | To create sensory interactions in street spaces | Allowing people to interact with public spaces and creating enrichment value | Social network analysis, wearable technology, VR |
| Protection of Vulnerable People | To provide a channel to call for help in case of disasters and crimes | Improving the coverage of the surveillance system, enhancing the convenience of calling for help, and optimizing alarm analysis | Thermal sensing device, machine dialogue, cloud platform |
| Environmental Monitoring | To monitor the street environment and collect data | Equipping urban furniture with low-carbon and energy-saving equipment and realizing multisource information collection and environmental self-assessment | Weather probes, noise sensors, solar panels, information terminals, automatic irrigation sensors |

(1) Smart transportation technology focuses on the scheduling of multiple types of vehicles via traffic flow data and feedback to the platform. Based on the collection of traffic flow data, smart technologies can create an urban green-wave transportation and transport hub and build a static traffic guidance system.

(2) In terms of convenient living, smart technology recommends the installation of multisource, interactive equipment in the urban furniture and advocates for more combined and functional furniture and facilities compatible with the necessities of everyday life, and it strives to handle daily business at the office building or home at any time.

(3)  The vital improvement of street space is created via art installations that employ listening, seeing, smelling, and tactile elements. Their wireless networks could collect interactive information from more comprehensive sources and use the portrait to propose targeted strategies.

(4)  For the protection of the vulnerable, smart technology can be implemented via thermal-sensing prompts and road studs at crossing facilities, and convenient urban furniture can be set up so vulnerable people can call for help, relying on real-time alarm-system monitoring and a one-button alarm device and by using tracking sensors to provide more comprehensive alarm information.

(5)  The collection of environmental monitoring information is mainly based on detection and interactive sensors. Smart vehicles have been designed on the principles of low carbon emissions and convenience. These collect and upload urban environmental data to the platform for analysis and use touchscreen media to achieve timely feedback.

### 3.3. Smart Street Management and Control Platform

The Smart Street Management and Control Platform (SSMCP) (Figure 7) consists of four layers: the basic information layer, technology platform layer, scene application layer, and institutional protection layer. The selected cases presented valuable techniques, development trends, and space demands, which could contribute to SSMCP development. For example, Figures 2–4 are linked to the basic information layer and the scene application layer, which propose methods to acquire data, equip sensors, interact with people, and apply technologies. Figure 5 is linked to the technology platform layer, which presents the idea that a comprehensive information platform should be built to serve the whole city; the platform should be consistent, allowing some technologies to be supported and some functions to be realized. Figure 6 is linked to basic information layer, which shows some methods to record urban statistics as urban basic information.

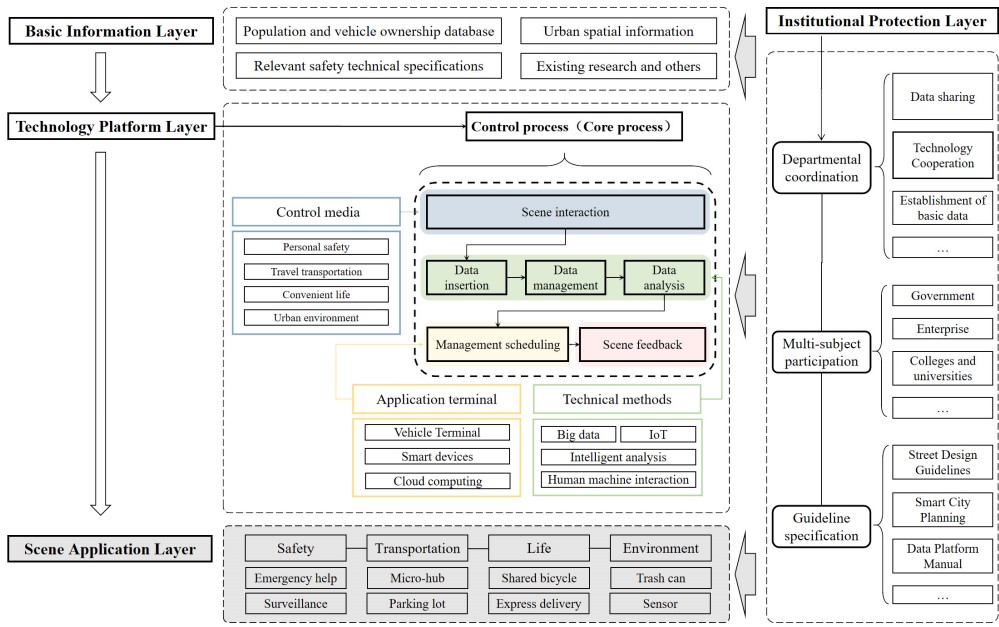

**Figure 7.** The SSMCP for chinese cities. Source: by author.

### 3.3.1. Basic Information Layer

The basic information layer collects data maps from multiple departments. The project's planning–construction–management process (whole-life cycle) must gradually promote linkages among departments, which can improve the efficiency of interdepartmental work in practice. A population database and vehicle ownership data are collected

and updated based on the urban spatial information census data in this layer. Based on the relevant exceptional safety technical specifications and existing research, this layer enriches data lists and supports subsequent technical operations. The improvement of data collection provides strong support for city managers to make decisions. In practice, the collection of data can rely on various types of sensors equipped in the streets, and new methods can be applied to achieve it. As presented for previous SDGs, urban data can be captured from basic surveys, transportation statistics, and records on public services and the environment. Among these, radar, photography, and portrait technologies can contribute to measuring various types of basic information.

### 3.3.2. Technology Platform Layer

The technology platform layer is the core layer of the whole construction. Smart technologies in the street space are the carriers of information processing and exchange, as well as sharing and collaboration. The control media consider four scenarios: personal safety, travel transportation, life convenience, and the urban environment, which are scenarios for interactive responses and data collection in daily life. The data accessed from control media, together with the basic information from the upper layer, are sent to the scene interaction for a series of operations including data insertion, data management, and data analysis. These three operational processes are supported by big data, IoT, GPS, cloud computing, wearable technology, and other technical devices. According to the actual situation of each city, it is important to set different goals in management scheduling to achieve efficient operation on the streets. Based on the results produced by the above process, they are linked to the next layer in the scene feedback for presentation. In general, the information generated by residents' participation in street activities is calculated in the corresponding media. With the basic information layer as a reference, the data are accessed, managed, and analyzed through the smart analytic platform. This layer can conduct analysis, automatically identify abnormalities, and realize the supervision of data. It is also the pre-processing stage of the scene application layer.

### 3.3.3. Institutional Protection Layer

The institutional protection layer guarantees the structure of the SSMCP from the perspectives of departmental coordination and multisubject participation. This layer is promoted by compiling corresponding documents, which also clarify the authority–responsiveness relationship and define the implementation and maintenance subjects for each layer of the platform. This layer establishes a multi-departmental and whole-process guarantee mechanism and improves the protection of specification documents, which in turn realize the operation and management of the whole SSMCP, and, finally, it achieves the goals of data collection, platform processing, rapid response, and collaborative feedback. Departmental coordination enables data sharing, technological cooperation, and the establishment of data framework, converged government, enterprise, and education institutions to achieve multi-subject participation, and enriches the types of guideline manuals used in academic research.

### 3.3.4. Scene Application Layer

Based on scenario feedback from the upper layer, the activities occurring in the street, the needs and possibilities of the current operation, and the final effects of the feedback are all expressed in the scene application layer. Consistent with the scenario types accessed from the control media, this layer presents the results through the same scenarios: safety, transportation, life, and environment. Security is represented by safety reminders and calling facilities in all kinds of urban furniture. The efficiency of mobility is optimized through the dispatching of vehicles and the prediction of traffic information. People can improve their quality of life based on the services they need. Feedback on the environment optimizes the efficiency of sanitation, greening, and other related departments and promotes sustainable development. This layer allows various smart technologies to be demonstrated,

and residents are able to experience how the scenarios of safety, transportation, life, and environment are realized in the streets. Through the residents' interactive experience, all kinds of facilities are connected to smart technologies and daily needs. Therefore, this layer can provide the most realistic picture of the SSMCP after being implemented in the street space. This layer could also visually demonstrate how the residents access a variety of smart technologies and how they fulfill their needs more sustainably.

## 4. Conclusions

Previous studies about smart cities have tended to be theoretical research, with less exploration of real cases. This paper studied the smart technologies of the street space. By sorting out the applications of smart technologies in several SDGs for five cities in China, the applications were categorized into five areas: smart transportation, convenient living, life enrichment, the protection of vulnerable people, and environmental monitoring. Then, to optimize the application of smart technologies, the SSMCP was explored. The SSMCP compounds the three functions of monitoring, controlling, and serving streets to create smart, efficient, vibrant, and safe streets. A response mechanism was constructed to enable data interaction, platform processing, and terminal feedback via four media, which included safety, transportation, life, and environment. Under the institutional protection of the system, the platform can improve the efficiency of data sharing and business collaboration among departments and enhance the sustainability and intelligence of the processes of "urban planning–construction–management" and holistic service. Smart technologies were used in the SSMCP to achieve street control, environmental management, and greater livability.

This conceptional and comprehensive framework for street space provides smart cities projects an actionable case and lays a foundation for future smart city advancement, which can be seen as a sample of smart city construction in practice, and can provide an original idea for a real street management platform in the future. The SSMCP could contribute to stock-based renewal and sustainable development, and the study could provide a reference for the implementation of smart cities in street space.

Because the selected cities were limited, it was not possible to summarize all the SDGs in China and all types of smart technologies according to region, and there was no section to illustrate how the SSMCP works in a certain city. Thus, in future studies, it is necessary to take more areas into account, to expand the scope of study, and to consider a practical example to explain the details.

**Supplementary Materials:** The following supporting information can be downloaded at: https://www.mdpi.com/article/10.3390/su15043438/s1, Table S1: The applications of smart technologies in the Beijing SDGs; Table S2: The applications of smart technologies in the Shanghai SDGs; Table S3: The applications of smart technologies in the Shenzhen SDGs; Table S4: The applications of smart technologies in the Nanjing SDGs; Table S5: The applications of smart technologies in the Qingdao SDGs.

**Author Contributions:** Conceptualization and methodology, F.X. and H.C.; data curation and investigation, F.X.; writing—original draft preparation, F.X.; writing—review and editing, H.C. and Y.W.; supervision and funding acquisition, H.C. All authors have read and agreed to the published version of the manuscript.

**Funding:** This research was funded by the National Natural Science Foundation of China, grant number 52170174.

**Institutional Review Board Statement:** Not applicable.

**Informed Consent Statement:** Not applicable.

**Data Availability Statement:** Not applicable.

**Conflicts of Interest:** The authors declare no conflict of interest.

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
