# Peer review of "Exploring the Smart Street Management and Control Platform from the Perspective of Sustainability: A Study of Five Typical Chinese Cities"

_sustainability, doi:10.3390/su15043438_

Round 1

Reviewer 1 Report

This is an interesting paper that discusses the framework of SSMCP(Smart Street Management of Control Platform) of China and uses five cities as case study. Here are the suggestions for revision.

In the abstract part, The background is overemphasized in the abstract section. L8-12, The abstract needs to be rewritten. Also, some words should be double-checked, such as L8, stock development, Is it a synonym for redevelopment or urban renewal?

L2 five major cities? Or five cities?

L8 What is “urban work”? ……entered a new phase of high-quality and sustainable development?

L23 In the keywords part, stock renewal may be not suitable.

L26-55, The background of smart cities in the world should be mentioned in the introduction section, as the gap of this research is not very clear. Why are you conducting this research? What is the significance of this study? Please elaborate.

L68-117 The literature review should be combined with the introduction. And, in terms of data, how many smart cities are there in China? Why did you select five cities? Figure 1 should be supplemented with the spatial location of five cities, which should be combined with table 1.

All the tables and figures should be clarified with the source of the data.

Figure 7 is not clear.

The conclusion just merely based on the case study without critically compare or referring to the previous findings.

References need to supplement in the revised vision. Each reference in the References section should be double-checked to ensure that it is correct.

Author Response

  • “The background of the abstract part is overemphasized and L8-12 needs to be rewritten. Is the stock development a synonym for redevelopment or urban renewal?”

Response: Thanks very much for the suggestions. First, we rewrote to simplify the background of abstract part and clarified the aim of this study as well. Second, we revised “stock development” to “stock-based development”. The “stock-based development” is a term of disciplinary of urban plan in China, which means do not merely make the land large-scale sprawl to get the economic benefits, but adopt a way to change the investment-driven model, and improve the urban quality. China is currently at this development stage, which stated that urban development should emphasize from the speed of economic growth to the construction of a livable city with people.

  • “L2 five major cities? Or five cities?"

Response: Thanks very much for the comment. We revised the word “major” into “typical”. These five cities we selected are representative in terms of the visibility of the city, the level of urbanization, the level of economy, and the guiding significance of the SDGs. Thus, the word “typical” should be more suitable.

  • “L8 What is “urban work”? ……entered a new phase of high-quality and sustainable development? “

Response: Thanks very much for the comment. The “urban work” is a term of disciplinary of urban plan in China, which means various types of things that needed to be deployed to promote urban construction and development. In the revised version, we revised “work” to “renewal” to make this sentence more accurate.

  • L23 In the keywords part, stock renewal may be not suitable.”

Response: Thanks very much for the comment. We revised “stock renewal” to “stock-based renewal”. Stock-based renewal mode is currently a mainstream renewal direction of urban planning in China, and preferred to conduct this mode in highly developed cities, like Beijing, Shanghai, Shenzhen, et al. . The results of this study proposed a way to improve the quality, efficiency and safety of urban space and public transportation. The way can be categorized to stock-based renewal mode.

  • “L26-55, The background of smart cities in the world should be mentioned in the introduction section, as the gap of this research is not very clear. Why are you conducting this research? What is the significance of this study? Please elaborate. “

Response: Thanks very much for the comment. We added sentences to clarify the background of smart cities in the world and noted the gap of this study. The reason why we conducted this research was that we wanted to set a sample of smart city construction in practical, which could provide origin idea of the street management platform in the future. Because in nowadays, Chinese experts quite discussed a lot about the possibility, feasibility of practical projects of smart cities, however, there was still not a proven and complete operational framework has been proposed. Especially in the disciplinary of urban planning, it was essential to propose a comprehensive system to server the whole city. Thus, the presentation of the SSMCP was the significance of this study.

  • L68-117 The literature review should be combined with the introduction. And, in terms of data, how many smart cities are there in China? Why did you select five cities? Figure 1 should be supplemented with the spatial location of five cities, which should be combined with table 1.”

Response: Thanks very much for the comment.

L68-117 is not the literature review section, but the experimental methods and data section. Coincidentally, there is a literature review method which was used in this study, and these two terms look so similar.

So far, Chinese government has announced three batches of smart city pilot projects, totaling 290 cities. And those cities have been making various attempts to try to consider integrating the concept of smart cities in economy, living environment, ecological regulation, disaster warning and cultural synergy development, and establish projects to practice the concept in residential areas, new urban districts, parks and other places.

The reason why we selected five cities was these five cities were typical and representative of the content of the SDGs, which proposed a series of proven and effective guidelines for the renewal of street space. Furthermore, the geographical distribution of these five cities reflected the overall situation in China.

We added the figure with the spatial location of five cities in the revised version.

  • “All the tables and figures should be clarified with the source of the data.”

Response: Thanks very much for the comment. We added notifications to clarify the source of the date of all the tables and figures in the revised version.

  • “Figure 7 is not clear.”

Response: Thanks very much for the comment. Figure 7 is a framework for street control and management, and this framework was developed based on the discipline of urban plan, so we added some new sentences to explain figure 7, especially to clarify how the four induvial layers and the whole process work in practical operation.

  • “The conclusion just merely based on the case study without critically compare or referring to the previous findings.”

Response: Thanks very much for the comment. We added new sentences in conclusion section to express the innovation and progress of this paper compared to previous study in the revised version. In previous study about smart cities, it tended to be theoretical research with less exploration of real cases. The result of this paper was kind of empirical studies, which was progressive compared to previous findings.

  • “References need to supplement in the revised vision. Each reference in the References section should be double-checked to ensure that it is correct.”

Response: Thanks very much for the comment. We have double-checked all references and revised the formats and mistakes.

Reviewer 2 Report

This paper analyzed the SDGs of five major Chinese cities from the lens of sustainability. It could contribute to the understanding of smart street management. My main concern is the research framework.

Sustainability is the balance between economic development, environmental protection, and social equity. Authors used five scenarios to analyze the SDGs, including smart transportation, convenient living, life enrichment, protection of vulnerable people. But how were the five criteria selected? If it is a sustainability framework, it misses the perspective of economic development. Also, not all the SDGs are analyzed using the same framework.

In each city, authors included a variety of information. There is not a holistic and consistence framework to justify the analysis. For example, authors emphasized the urban furniture in Beijing, the integration of street space in Shanghai, smart life micro-hub in Shenzhen, the information platform of Nanjing, and the business activity measurement portrait system in Qingdao. The figures are helpful, not are not well explained. Also, it is not clear that how those figures are linked to the framework.

In the section 3.3, authors descripted four layers of the SSMCP. How were the layers developed? There are not any studies cited in the section. Figure 7 is very complex and is not explained clearly. If Figure 7 is the framework, the case studies should focus on how those four layers work in each city. But, none of the layers were discussed.

What are the limitations of this study?

To sum up, this paper provides detailed information on the SDGs of five major Chinese cities. More work is needed to refine the research framework. All the sections need to be linked. Now, the paper reads more like a research report. A conceptional framework/research framework at the front section could be helpful.

Author Response

  • “Sustainability is the balance between economic development, environmental protection, and social equity. Authors used five scenarios to analyze the SDGs, including smart transportation, convenient living, life enrichment, protection of vulnerable people. But how were the five criteria selected? If it is a sustainability framework, it misses the perspective of economic development. Also, not all the SDGs are analyzed using the same framework.”

Response: Thanks very much for the comment.

From the beginning, we analyzed the smart technologies from those SDGs. We wanted to know that how many kinds of smart technologies SDGs had put forward, as well as how SDGs suggested applying these smart technologies and where to. Then we found that different city made different suggestions. Some had discussed the application of traffic assistance, facilities interaction and environmental monitoring, which discussed more application scenarios. Some had discussed the application of space intensification, safety protection, and so on, which discussed fewer. So we did not use the same framework to analyze the SDGs. We concluded five tables in supplementary material to record the origin expression from the five SDGs. Finally, as a contrast, we compared the application of smart technologies of five cities and distilled the five scenarios, which were smart transportation, convenient living, life enrichment, protection of vulnerable people and environmental monitoring. ,

The reason why we did not mention the economic development was that all the selected SDGs had not discussed this topic. The SDGs just proposed various operations that were easy to implement in the street space renewal, and did not involve some discussion at the macro-economic level.

  • “In each city, authors included a variety of information. There is not a holistic and consistence framework to justify the analysis. For example, authors emphasized the urban furniture in Beijing, the integration of street space in Shanghai, smart life micro-hub in Shenzhen, the information platform of Nanjing, and the business activity measurement portrait system in Qingdao. The figures are helpful, not are not well explained. Also, it is not clear that how those figures are linked to the framework.”

Response: Thanks very much for the comment. As we began the process of analyzing the SDGs, we found that most cities had some insightful directions. For example, the Beijing SDGs had suggested integrating sensors in urban furniture to save street space, providing more public service and collecting street data in the while, the Shenzhen SDGs had suggested setting a life micro-hub to make people with different roles in the city to get corresponding public service, likewise the SDGs in other cities also presented some technical methods, some development trends or demands. So, we felt it was necessary to record those important details (Figure 2-6), and these details are an important theoretical basis for the proposed framework (Figure 7). We revised the chapter 3.3 with the link of figures and the platform. For example, Figure 2-4 were linked to Basic Information Layer and Scene Application Layer, which proposed methods to acquiring data, locations of sensors, interactive with people and the application of technologies. Figure 5 was linked to Technology Platform Layer, which presented an idea that a comprehensive information platform should be built to serve the whole city, the platform should be consist with some technologies to supported and some functions to realize. Figure 6 was linked to Basic Information Layer, which showed some methods to record the urban statistics as urban basic information.

  • “In the section 3.3, authors descripted four layers of the SSMCP. How were the layers developed? There are not any studies cited in the section. Figure 7 is very complex and is not explained clearly. If Figure 7 is the framework, the case studies should focus on how those four layers work in each city. But, none of the layers were discussed.”

Response: Thanks very much for the comment. We added some sentences to clarify how the four induvial layers and the whole process developed in practical operation in revised version. Generally speaking, we sorted out the smart technologies of the SDGs at first, and concluded five application scenarios. Then, we recorded important details of five SDGs. Some reveled the real demand in the urban space, some showed the theoretical basis, and some provided supplementary function of street. Based on the above study, tables and figures were developed.

The relationship between selected cases and SSMCP was clarified in the revised version. We added sentences to clarify the Figure 7. The biggest highlight of this paper was to propose a control and management platform to realize sustainable high-quality development based on street renewal, so we focused more on the formation process of SSMCP, and specific details about the operation of SSMCP would be paid more attention in the future study.

  • “What are the limitations of this study?”

Response: Thanks very much for the comment. We supplemented the limits in conclusion part. The limitations could be stated as follows.

The first limitation is that we only selected five SDGs to study, the sample size is insufficient. After the cities in western and northern area of China compiled SDGs, we should take it into account to expand the scope of study. The second limitation is that we did not use a specific city as an example to illustrate the SSMCP how they developed, and this could be completed in the next step.

Round 2

Reviewer 1 Report

The authors make a good effort to rectify this paper better.

Reviewer 2 Report

I appreciated authors' efforts on revising the paper. The paper is much clearer now.